# Identifying Influencing Factors of Agricultural Soil Heavy Metals Using a Geographical Detector: A Case Study in Shunyi District, China

**Shiwei Dong** [1], **Yuchun Pan** [1], **Hui Guo** [2], **Bingbo Gao** [3] **and Mengmeng Li** [4,*]

1   Information Technology Research Center, Beijing Academy of Agriculture and Forestry Sciences, Beijing 100097, China; dongsw@nercita.org.cn (S.D.); panyc@nercita.org.cn (Y.P.)
2   Forestry Experiment Center of North China, Chinese Academy of Forestry, Beijing 102300, China; guohuistz@caf.ac.cn
3   College of Land Science and Technology, China Agricultural University, Beijing 100193, China; gaobingbo@cau.edu.cn
4   Key Lab of Spatial Data Mining & Information Sharing of Ministry of Education, Academy of Digital China (Fujian), Fuzhou University, Fuzhou 350108, China
*   Correspondence: mli@fzu.edu.cn

**Abstract:** Identifying influencing factors of heavy metals is essential for soil evaluation and protection. This study investigates the use of a geographical detector to identify influencing factors of agricultural soil heavy metals from natural and anthropogenic aspects. We focused on six variables of soil heavy metals, i.e., As, Cd, Hg, Cu, Pb, Zn, and four influencing factors, i.e., soil properties (soil type and soil texture), digital elevation model (DEM), land use, and annual deposition fluxes. Experiments were conducted in Shunyi District, China. We studied the spatial correlations between variables of soil heavy metals and influencing factors at both single-object and multi-object levels. A geographical detector was directly used at the single-object level, while principal component analysis (PCA) and geographical detector were sequentially integrated at the multi-object level to identify influencing factors of heavy metals. Results showed that the concentrations of Cd, Cu, and Zn were mainly influenced by DEM ($p = 0.008$) and land use ($p = 0.033$) factors, while annual deposition fluxes were the main factors of the concentrations of Hg, Cd, and Pb ($p = 0.000$). Moreover, the concentration of As was primarily influenced by soil properties ($p = 0.026$), DEM ($p = 0.000$), and annual deposition flux ($p = 0.000$). The multi-object identification results between heavy metals and influencing factors included single object identification in this study. Compared with the results using the PCA and correlation analysis (CA) methods, the identification method developed at different levels can identify much more influencing factors of heavy metals. Due to its promising performance, identification at different levels can be widely employed for soil protection and pollution restoration.

**Keywords:** soil sample; natural and anthropogenic factors; identification; multi-object; spatial analysis; agricultural land; principal component analysis

## 1. Introduction

Heavy metals in agricultural soil can reduce the soil quality, affect the health of crops, and even threaten human health [1,2]. Along with the progress of modern agriculture and industry, soil environmental problems such as soil pollution, soil erosion, and degradation gradually emerged in many areas [3,4]. To address these problems, it is essential to identify the influencing factors of heavy metals in agricultural soil for soil evaluation and protection. The influencing factors of heavy metals are usually grouped into two main types, corresponding to natural and anthropogenic factors. Natural factors mainly refer to soil properties (e.g., soil type, soil texture) [5], digital elevation model (DEM) or terrain [6], and distance from water body [7], while anthropogenic factors usually refer to land use [8,9], water irrigation, traffic emission [10] and atmospheric deposition [11]. The concentrations

of heavy metals in agricultural soil were attributed to the combination of natural and anthropogenic factors [12–14]. Due to the different mobility and availability of heavy metals, both natural and anthropogenic factors of each heavy metal concentration need to be determined. It is thus essential to develop an effective method to identify influencing factors for multiple heavy metals in agricultural soil.

Existing methods to identify influencing factors of heavy metals are generally based upon multivariate analysis, geostatistical, and geographical detector methods. Multivariate analysis methods such as principal component analysis (PCA) [15], correlation analysis (CA), cluster analysis, and geographically weighted regression (GWR) [16] rely on the distribution characteristics of heavy metals elements and can estimate heavy metal enrichment with multiple influencing factors. However, these methods neglect the spatial relationship between influencing factors and heavy metal concentrations. PCA was a frequently used method to identify natural and anthropogenic influencing factors of agricultural soil heavy metals in the past [17]. PCA is a multivariate statistical technique used to reduce the number of variables in a data set into a set of values of uncorrelated principal components using an orthogonal transformation [18]. Each component is a linear weighted combination of the initial variables, and the first component contains the most variance, followed by subsequent components. Principal components loadings show the weights or influence of individual elements in each component, where larger positive and negative values are of equal importance. Geostatistical methods, e.g., spatial interpolation, spatial mapping, and hot spots analysis [19,20], can analyze the contribution of different factors to the spatial distribution characteristics. However, they fail to measure the influencing degree of each specific factor quantitatively. Moreover, these methods require a relatively high number of samples for statistical inference [21].

An alternative is to use a geographical detector [22,23], which indicates that heavy metal exhibits a spatial distribution and spatial variability similar to that of the influencing factor [24]. It can quantitatively examine the spatial relationship between heavy metals and influencing factors. The geographical detector method was first applied to study neural tube defects in 2010 [22]. It has been widely used in soil science [25,26], public health [27,28], and other fields in recent years. Existing contributions to the topic of soil heavy metals and geographical detectors are grouped into three main subjects—namely, source identification [29,30], risk assessment [31,32], and spatial pattern analysis [33–35] of heavy metals. Most of these studies were located in China, with agricultural soils, forestland, and urban soils, and focused on analyzing multiple heavy metals, especially for Cd and Hg [29]. For analytical and statistical methods, a geographical detector was directly used to analyze the relationships between soil heavy metals and related factors, and corresponding significant levels were customized as 0.001 [31], 0.01 [32], and 0.05 [35], respectively. The geographical detector method was compared to other frequently used methods (e.g., PCA) to assess the advantages and disadvantages of the geographical detector method to other methods. Inspired by these studies, integrating PCA and geographical detectors may improve the identification of influencing factors of heavy metals.

This study aims to identify influencing factors of agricultural soil heavy metals in Shunyi District, China, using a geographical detector. We focused on identifying natural factors (e.g., soil properties and DEM) and anthropogenic factors (e.g., land use and annual deposition fluxes) of six heavy metals (including As, Cd, Hg, Cu, Pb, and Zn) at single-object and multi-object levels, and compared with the frequently used PCA method. Previous studies directly used a PCA or geographical detector to identify influencing factors of heavy metals and lacked corresponding systematic and comprehensive solutions at different levels. In this study, we directly used a geographical detector to identify influencing factors of heavy metals at a single-object level, and then integrated PCA and the geographical detector to identify these factors at the multi-object level.

## 2. Materials and Methods

### 2.1. Study Area

The study area is located at Shunyi District, Beijing, China, at longitudes ranging from 116°28′ E to 116°58′ E and latitudes ranging from 40°00′ N to 40°18′ N (Figure 1). This area covers 1020 km² and has a warm, temperate semi-humid continental monsoon climate [36], with an average annual temperature of 11.5 °C and average precipitation of 625 mm. The DEM changes from 24 m to 637 m in the area, and the average elevation is about 35 m. The primary agricultural lands in the area are irrigable land, orchard, vegetable field, grassland, and wasteland. Heavy metal concentrations in Shunyi District were slightly higher than the background values of Beijing topsoils except for Pb, suggesting slight contamination of heavy metals in this area [17].

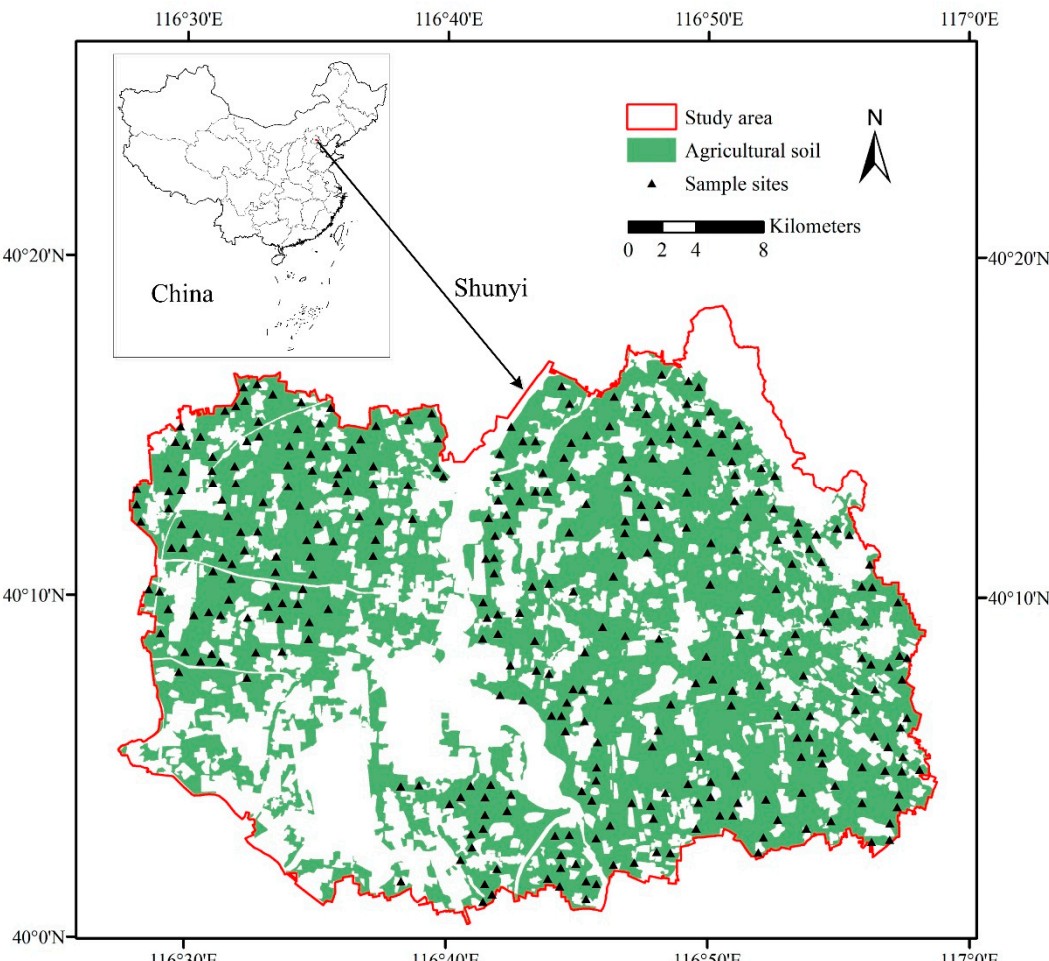

**Figure 1.** Location of the study area, Shunyi District, China.

### 2.2. Data Sources

#### 2.2.1. Sample Collection and Analysis

To investigate the pollution status of heavy metals in Shunyi District, a large-scale soil sampling project was conducted after the crop harvest in the autumn of 2007. A total of 329 surface soil samples (0–20 cm depth) were collected from the agricultural areas, as shown in Figure 1. For each sampling site, five sub-samples were collected from the four vertexes and the center of a 10 m × 10 m grid. We then mixed these sub-samples to select 1 kg soil as the representative sample of this site. A global positioning system (GPS) was used to precisely locate each sampling location and record the corresponding information regarding vegetation and soil types.

The collected soil samples were air-dried, crushed in an agate mortar, and then passed through a 2.0 mm sieve. The concentrations of six heavy metals, i.e., Cd, Cu, Zn, Pb, As, and Hg, were analyzed in the soil samples. The soil samples were digested in triplicate with the mixture of $HNO_3$, HCI, and $H_2O_2$ using the method 3050B recommended by the United States Environmental Protection Agency [37]. Concentrations of Cd, Cu, Pb, and Zn in the digestion solution were determined by inductively coupled plasma optical emission spectroscopy (ICP-AES, Thermo iCAP 6300, Washington, USA). In contrast, concentrations of As and Hg in the soils were determined by an atomic fluorescence spectrometry (Titan, AFS 830, Beijing, China) after digestion of the soil samples. Standard reference materials (GSS-1 and GSS-4) obtained from the Center of National Standard Reference Material of China were used for quality assurance and quality control [17]. Figure 2 shows the spatial distribution of each heavy metal of Shunyi District, China. In the distribution maps, the concentrations of each heavy metal were divided into 3 classes using the quantile method, which placed an equal number of units into each class [38] and seemed to be one of the effective methods for facilitating comparison [39].

2.2.2. Influencing Factors and Factors Stratification

This study examined four factors, including two natural and two anthropogenic factors. These factors were known to influence heavy metal concentrations in Shunyi District, China. The natural factors referred to soil properties, i.e., soil type and soil texture, and DEM. The two anthropogenic factors referred to land use and annual deposition fluxes of dry and wet atmospheric deposition. Soil properties data at a scale of 1:1,000,000 and DEM with a 30 m resolution were provided by the Institute of Geographic Sciences and Natural Resources Research, Chinese Academy of Sciences. The land use data at a scale of 1:50,000 were acquired from the National Administration of Surveying, Mapping, and Geoinformation. Annual deposition fluxes data of Cd, Cu, Zn, Pb, As, and Hg came from the inverse distance weight (IDW) [40] interpolation results of 39 samples of dry and wet atmospheric deposition, which were collected in the plain area of Beijing from November 2005 to November 2006 [41].

Factors stratification for corresponding influencing factors of 329 soil samples was used in this study. Factors stratification ensures that the spatial heterogeneity of soil pollutants can be identified after stratification [42]. The principle of factors stratification was that every stratum had sampling sites, and each area of different strata was similar. For category variables, e.g., soil properties and land use types, each category can be considered a stratum. For numerical variables, e.g., DEM and annual deposition fluxes, the variable was divided into three grades referring to the high, medium, and low levels using the quantile method, and each grade can be considered as a stratum. In this study, agricultural lands in Shunyi District were divided into four strata, corresponding to four land use types: irrigable land, orchard, vegetable field, and grassland and wasteland. Soil properties (soil type and soil texture) were divided into six strata, i.e., cinnamon soils and light loam, cinnamon soils and medium loam, cinnamon soils and sandy loam, fluvo-aquic soils and light loam, fluvo-aquic soils and medium loam, and fluvo-aquic soils and sandy loam. DEM and annual deposition fluxes were divided into three strata using the quantile method, i.e., low, medium, and high grades, respectively, and the stratification results are shown in Table 1.

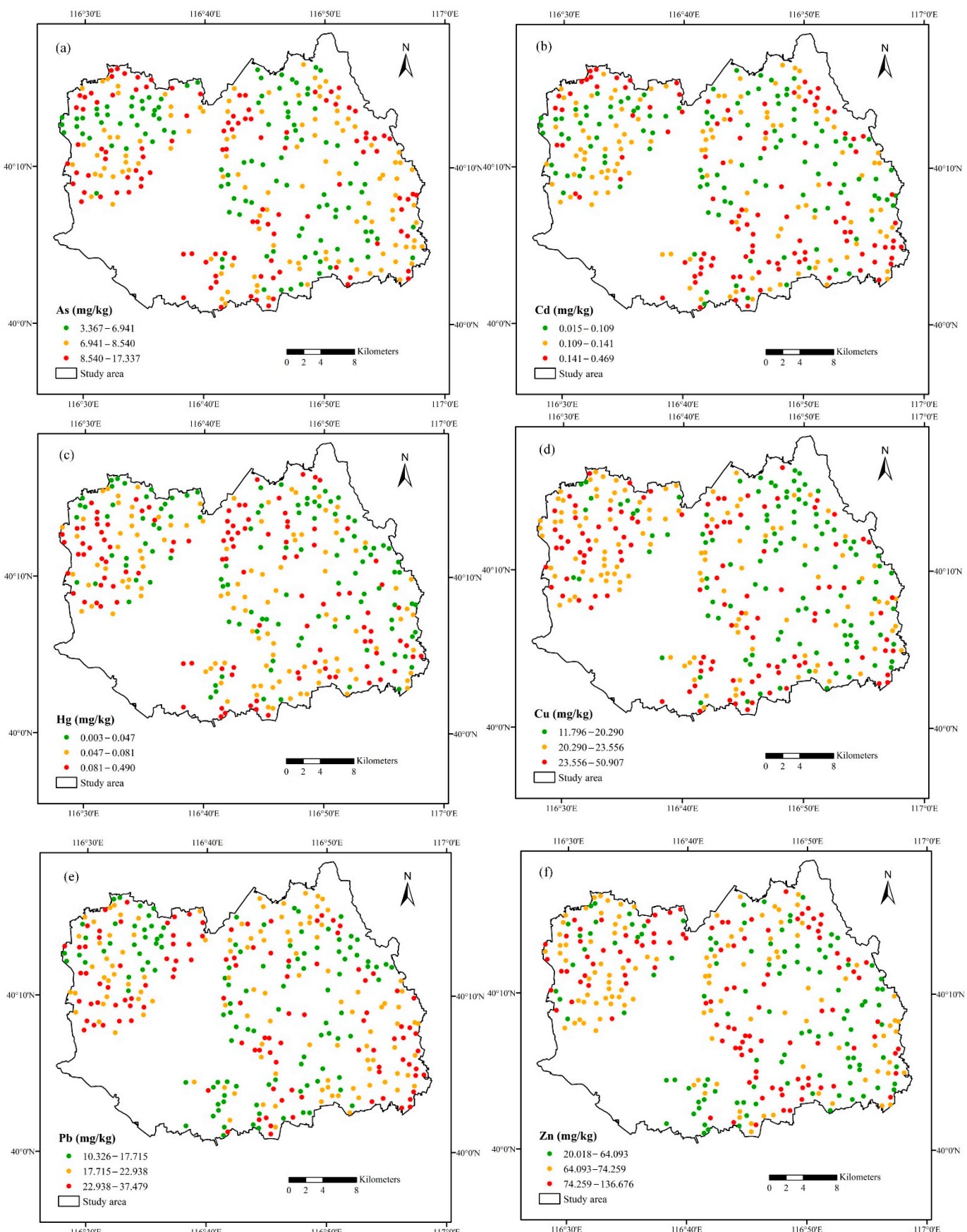

**Figure 2.** Spatial distribution of As (**a**), Cd (**b**), Hg (**c**), Cu (**d**), Pb (**e**), and Zn (**f**) of Shunyi District, China.

**Table 1.** The stratification of influencing factors for 329 soil samples.

| Influencing Factor | | Mean | Minimum | Maximum | Strata | | |
|---|---|---|---|---|---|---|---|
| | | | | | Low | Medium | High |
| DEM (m) | | 45.444 | 25.304 | 100.000 | <40.830 | 40.830–49.533 | >49.533 |
| Annual deposition fluxes (g/hm²·a) | As | 29.976 | 25.775 | 34.107 | <28.630 | 28.630–31.089 | >31.089 |
| | Cd | 1.878 | 1.320 | 2.940 | <1.658 | 1.658–1.967 | >1.967 |
| | Hg | 0.218 | 0.211 | 0.233 | <0.213 | 0.213–0.220 | >0.220 |
| | Cu | 141.086 | 134.035 | 147.863 | <140.794 | 140.794–142.290 | >142.290 |
| | Pb | 202.760 | 183.859 | 221.683 | <199.871 | 199.871–206.722 | >206.722 |
| | Zn | 503.260 | 484.701 | 544.997 | <493.103 | 493.103–506.092 | >506.092 |

### 2.3. Identification Method

In this study, we developed a method based upon geographical detector and PCA to identify natural and anthropogenic influencing factors of agricultural soil heavy metals, i.e., As, Cd, Hg, Cu, Pb, and Zn, at both single-object and multi-object levels, as depicted in Figure 3. We proposed a systematic and comprehensive solution at different levels to identify natural and anthropogenic influencing factors of agricultural soil heavy metals. At the single-object level, we directly used a geographical detector to identify natural factors or anthropogenic factors of each heavy metal, i.e., As, Cd, Hg, Cu, Pb, or Zn. At the multi-object level, we first used PCA to represent the concentrations of heavy metals including As, Cd, Hg, Cu, Pb, and Zn in agricultural soils. After PCA processing of As, Cd, Hg, Cu, Pb, and Zn, we selected principal components with eigenvalues greater than 1. Then, we used a geographical detector to identify natural factors or anthropogenic factors of each selected principal component. Furthermore, we obtained identification results at single-object and multi-object levels by $q$ values, with the significance at $p < 0.05$ level.

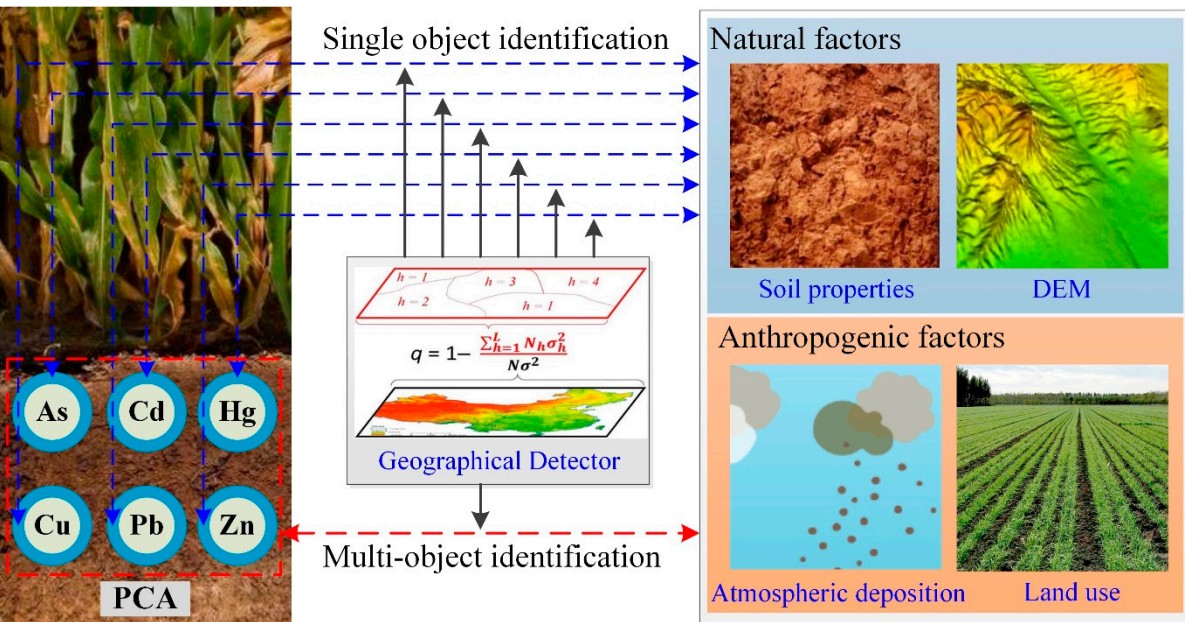

**Figure 3.** The proposed method for factor identification at a single-object and multi-object levels.

The geographical detector is a spatial statistical method to test the relationships between geographical phenomena and their potential influencing factors [22,23]. The

geographical detector quantifies the relative importance of an influencing factor on the heavy metal concentrations by power determinant (*q*). The *q* is defined as follows:

$$q = 1 - \frac{1}{N\sigma^2} \sum_{h=1}^{L} N_h \sigma_h^2 \tag{1}$$

where *h* = 1, 2, . . . , *L* indicates a stratum; *N* is the number of samples; $N_h$ is the number of strata. The parameter $\sigma^2$ refers to the variance of a heavy metal concentration for the whole study area, and $\sigma_h^2$ is the corresponding variance of a stratum. In this study, we used the significance of *q* value (*p*) as the key index to identify natural and anthropogenic influencing factors of heavy metals. The significance of the *q* value was defined at $p < 0.05$ level, and the corresponding *q* value indicated the influencing degree of this factor on heavy metals. For each natural and anthropogenic influencing factor of heavy metals, the *p* of the corresponding *q* value is less than the target significance (0.05), which indicates this influencing factor can be identified; otherwise, it cannot be identified.

In this study, we implemented the geographical detector algorithm using the software accessed from http://www.geodetector.cn/ (accessed on 15 October 2020). The CA, PCA, and data processing were carried out by SPSS 16.0 software. The fuzzy clustering was performed by MATLAB 7.0 software. The quantile method and spatial mapping were performed by ArcGIS 10.5 software.

## 3. Results

### 3.1. The Identification of Influencing Factors of Heavy Metals at a Single-Object Level

We directly identified natural and anthropogenic factors of each heavy metal, i.e., As, Cd, Hg, Cu, Pb, or Zn, in Shunyi District, China. According to Equation (1), the obtained *q* values and corresponding significance *p* of influencing factors on each heavy metal concentration at a single-object level are shown in Table 2. The *q* values with the significance at $p < 0.05$ level are illustrated in Figure 4. This figure showed that As, Cd, and Cu concentrations were mainly derived from DEM, while annual deposition fluxes primarily influenced Cd, Hg, and Pb concentrations in Shunyi District. Meanwhile, annual deposition flux had a much higher association than DEM for Cd concentrations, and Zn concentrations had no association with all influencing factors, as shown in Table 2 and Figure 4.

**Table 2.** The influencing factors on heavy metal concentrations at the single-object level.

| Metal | Index | Soil Properties | DEM | Land Use | Annual Deposition Fluxes | | | | | |
|---|---|---|---|---|---|---|---|---|---|---|
| | | | | | As | Cd | Hg | Cu | Pb | Zn |
| As | *q* | 0.073 | 0.093 | 0.024 | 0.008 | | | | | |
| | *p* | 0.418 | 0.000 | 0.457 | 0.282 | | | | | |
| Cd | *q* | 0.025 | 0.044 | 0.022 | | 0.100 | | | | |
| | *p* | 0.660 | 0.000 | 0.328 | | 0.000 | | | | |
| Hg | *q* | 0.028 | 0.012 | 0.012 | | | 0.040 | | | |
| | *p* | 0.150 | 0.144 | 0.363 | | | 0.000 | | | |
| Cu | *q* | 0.016 | 0.023 | 0.050 | | | | 0.013 | | |
| | *p* | 0.989 | 0.024 | 0.261 | | | | 0.134 | | |
| Pb | *q* | 0.015 | 0.011 | 0.004 | | | | | 0.022 | |
| | *p* | 0.989 | 0.183 | 0.977 | | | | | 0.028 | |
| Zn | *q* | 0.003 | 0.000 | 0.021 | | | | | | 0.008 |
| | *p* | 1.000 | 0.982 | 0.770 | | | | | | 0.259 |

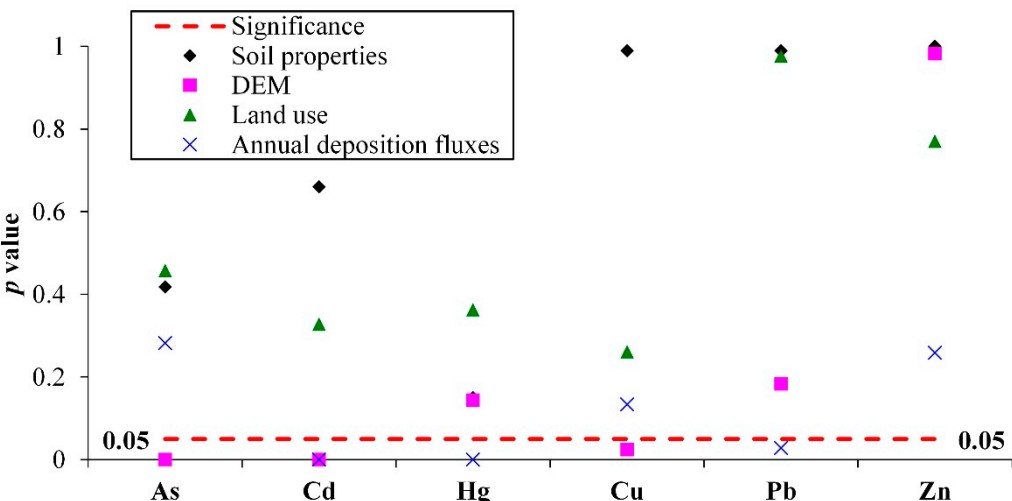

**Figure 4.** The *p* values of influencing factors on heavy metal concentrations at the single-object level.

*3.2. The Identification of Influencing Factors of Heavy Metals at the Multi-Object Level*

At the multi-object level, we first used PCA to show the concentrations of six heavy metals, i.e., As, Cd, Hg, Cu, Pb, and Zn, in agricultural soils for Shunyi District, China, and the PCA results are presented in Table 3. The eigenvalues of the first three extracted components were all greater than 1. The heavy metals in the area can be represented by the first three primary components that account for 64.496% of all the data variation. The component matrix showed that Cd, Cu, and Zn were strongly associated with the first component because of selecting the maximum principal component for each heavy metal. Moreover, As was the only element represented in the second component, while Hg and Pb mainly dominated the third component.

**Table 3.** PCA results for six heavy metals in Shunyi District.

| Component | Eigenvalues | | Component Matrix | | | | | |
|---|---|---|---|---|---|---|---|---|
| | Total | Cumulative % | As | Cd | Hg | Cu | Pb | Zn |
| 1 | 1.747 | 29.124 | 0.385 | 0.609 | 0.345 | 0.801 | 0.337 | 0.595 |
| 2 | 1.092 | 47.319 | 0.738 | 0.218 | −0.069 | −0.242 | 0.371 | −0.546 |
| 3 | 1.031 | 64.496 | −0.141 | 0.444 | 0.647 | −0.048 | −0.482 | −0.401 |
| 4 | 0.954 | 80.397 | | | | | | |
| 5 | 0.686 | 91.832 | | | | | | |
| 6 | 0.490 | 100.000 | | | | | | |

In the second stage, we identified influencing factors including soil properties, DEM, land use, and annual deposition fluxes of each of the first three principal components using geographical detector software, and the identification results are shown in Table 4. Through the *q* values with the significance at $p < 0.05$ level, the results in Table 4 and Figure 5 suggested that the first component represented Cd, Cu, and Zn concentrations, which were mainly derived from DEM, land use, and annual deposition fluxes of As, Cd, Hg, and Pb. The second component represented As concentrations, which were primarily influenced by soil properties, DEM, and the annual deposition flux of Cd. The third component represented Hg and Pb concentrations, which were mainly influenced by annual deposition fluxes of Cd and Hg.

**Table 4.** The influencing factors on heavy metal concentrations at the multi-object level.

| Component | Index | Soil Properties | DEM | Land Use | Annual Deposition Fluxes | | | | | |
|---|---|---|---|---|---|---|---|---|---|---|
| | | | | | As | Cd | Hg | Cu | Pb | Zn |
| 1 | $q$ | 0.025 | 0.031 | 0.028 | 0.038 | 0.041 | 0.025 | 0.002 | 0.033 | 0.014 |
| | $p$ | 0.170 | 0.008 | 0.033 | 0.003 | 0.000 | 0.018 | 0.709 | 0.005 | 0.106 |
| 2 | $q$ | 0.045 | 0.064 | 0.025 | 0.003 | 0.055 | 0.013 | 0.006 | 0.000 | 0.003 |
| | $p$ | 0.026 | 0.000 | 0.053 | 0.571 | 0.000 | 0.131 | 0.381 | 0.924 | 0.590 |
| 3 | $q$ | 0.016 | 0.013 | 0.012 | 0.002 | 0.051 | 0.070 | 0.004 | 0.008 | 0.005 |
| | $p$ | 0.424 | 0.126 | 0.294 | 0.722 | 0.000 | 0.000 | 0.490 | 0.286 | 0.441 |

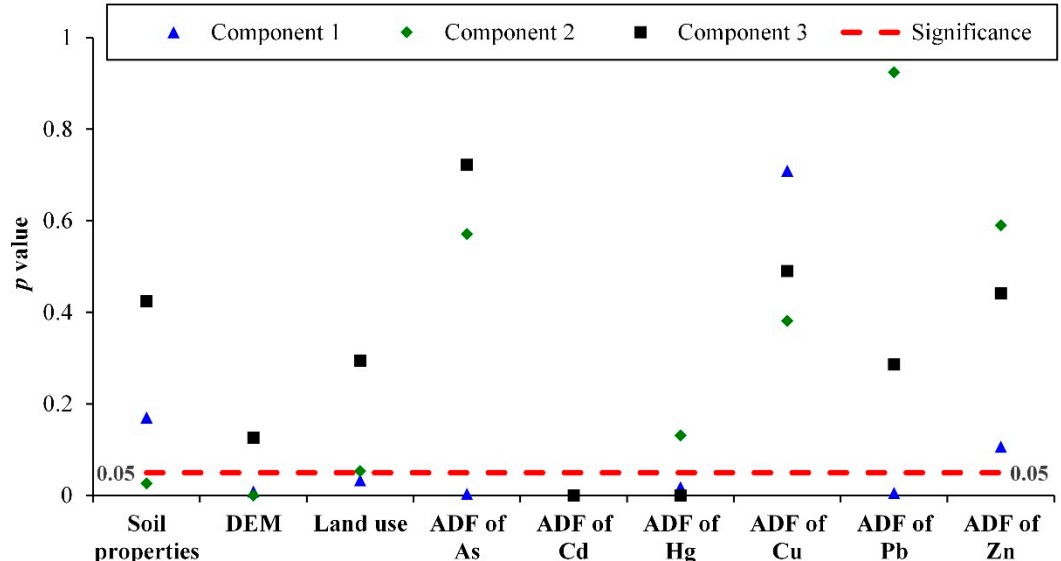

**Figure 5.** The *p* values of influencing factors on heavy metal concentrations at the multi-object level: ADF, annual deposition flux.

### 3.3. Comparative Analysis of Cluster Analysis and Correlation Analysis Methods

We performed a comparative analysis between the proposed method and existing methods using cluster and correlation analysis. First, a fuzzy clustering method was used to represent the concentrations of heavy metals including As, Cd, Hg, Cu, Pb, and Zn in agricultural soils. Figure 6 shows the results of the fuzzy clustering for six heavy metals in Shunyi District. Compared with the PCA results of heavy metals, we selected three strata of the cluster results for comparison. Six heavy metals were divided into three strata: Cd, Zn, and Cu (first stratum), Pb and As (second stratum), Hg (third stratum) using fuzzy clustering, while three principal components represented Cd, Cu, and Zn (first component), As (second component), Hg and Pb (third component) using PCA, respectively. The results of the fuzzy clustering method and PCA method were generally consistent except for Pb.

Second, we also conducted a correlation analysis between heavy metal concentrations and influencing factors (Table 5). DEM was correlated with As, Cd, Hg, and Cu, while land use was correlated with Cd, Cu, and Zn. Moreover, annual deposition fluxes were correlated with Cd and Hg. Figure 7 shows the identification results for multi-object identification using the proposed method and the CA method. According to Figure 7, the multi-object identification results between heavy metals and influencing factors derived using the proposed method contained those derived from the CA method, except for the variable of Hg.

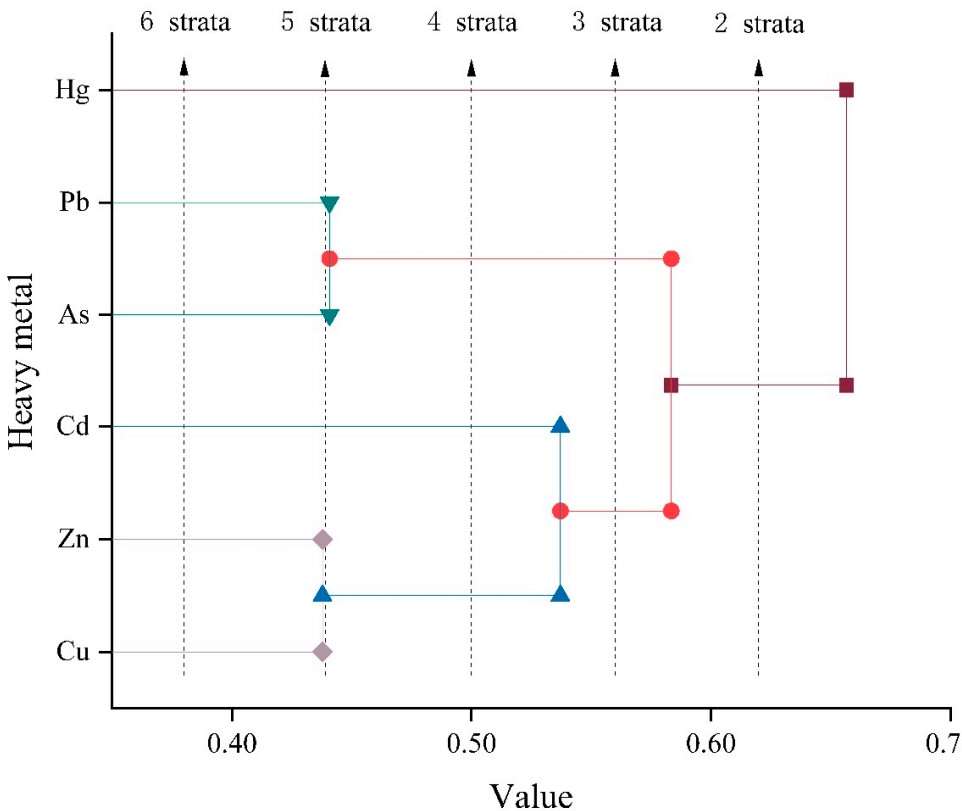

**Figure 6.** Fuzzy clustering results for six heavy metals in the Shunyi District.

**Table 5.** The correlation analysis results of heavy metal concentrations and influencing factors.

| Metal | Soil Properties | DEM | Land Use | Annual Deposition Fluxes | | | | | |
|---|---|---|---|---|---|---|---|---|---|
| | | | | As | Cd | Hg | Cu | Pb | Zn |
| As | 0.101 | 0.143 ** | 0.031 | 0.046 | | | | | |
| Cd | 0.079 | −0.150 ** | −0.109 * | | 0.282 ** | | | | |
| Hg | 0.064 | −0.110 * | 0.014 | | | 0.150 ** | | | |
| Cu | 0.105 | −0.153 ** | −0.205 ** | | | | 0.069 | | |
| Pb | 0.099 | −0.074 | 0.031 | | | | | 0.049 | |
| Zn | 0.024 | 0.004 | −0.142 * | | | | | | 0.019 |

** Correlation is significant at the 0.01 level (2-tailed); * correlation is significant at the 0.05 level (2-tailed).

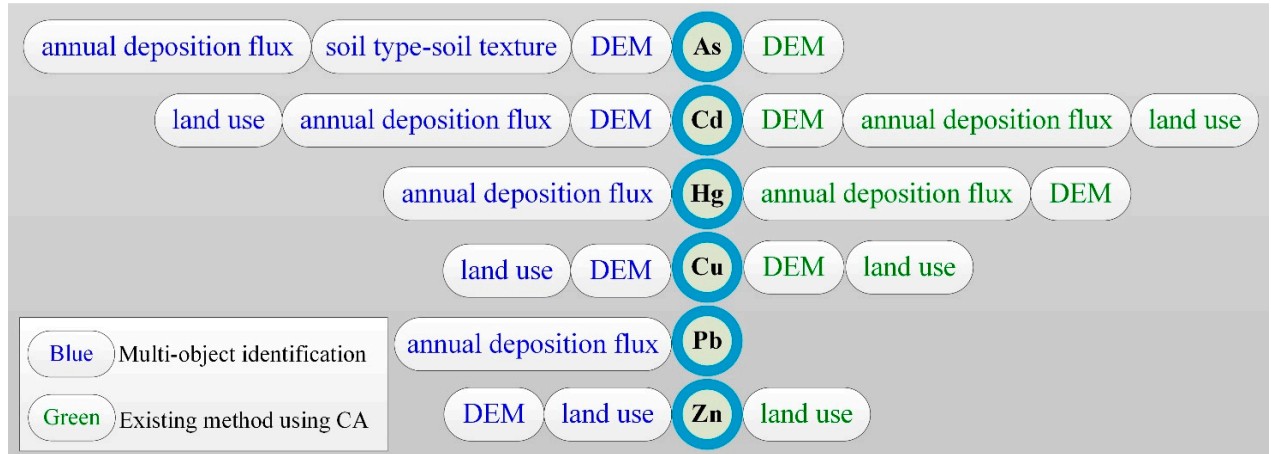

**Figure 7.** The identification results for multi-object identification and CA methods.

## 4. Discussion

### 4.1. Comparisons with Related Studies

For six kinds of heavy metals in agricultural soils, the influencing factors of their concentrations were mainly related to land use, annual deposition flux, DEM, soil type, and soil texture. This finding reflects that the concentrations of heavy metals in agricultural soil were attributed to the combination of natural and anthropogenic factors. Based upon existing research [17], local contamination from agricultural practices, particularly with the application of fertilizer and manure, was the main source of Cd, Cu, and Zn entering the agricultural soils. Their results support our finding that land use was the main influencing factor for Cd, Cu, and Zn in this study. The concentration of As was primarily influenced by soil properties, which was proved by the existing contributions [35,43]. Moreover, DEM was the secondary influencing factor for Cd and As [42], and atmospheric deposition played an important role in Cd, Hg, and Pb accumulations in the soils [17,32]. The identification results of heavy metal sources in agricultural soils are essential for protecting soil quality to promote the healthy development of agriculture.

To demonstrate the performance of the identification method at different levels in this study, we made two comparisons, i.e., between the single-object identification method and multi-object identification method, and between the identification method developed in this study and related studies [2,17,43–45]. The results of the comparative analysis of this study and existing contributions are shown in Table 6. Table 6 shows that the identification results of different studies were generally consistent in terms of the concentrations of heavy metals influenced by the combination of natural and anthropogenic factors. In contrast, the influencing factors of each heavy metal were different between different studies.

**Table 6.** The comparative results between this study and existing contributions.

| Reference | Time | Location | Type | Samples | Method | Metals | Influencing Factors | Significant |
|---|---|---|---|---|---|---|---|---|
| Our study | Autumn 2007 | Shunyi District | agricultural soils | 329 | Geographical Detector, PCA | As, Cd, Cu, Cd, Hg, Pb | land use, annual deposition flux, DEM, soil type, and soil texture | $p < 0.05$ |
| [29] | October 2018 | Fengcheng, Jiangxi, China | farmland soils | 283 | Geostatistics, Geodetector | Cd, Hg | soil pH, total phosphorous, elevation, distance from a river, distance from a road | $p < 0.05$ |
| [30] | July 2016, April 2017 | Guangxi | karst soils | 117 | Geographical Detector | Cd | soil type, geological age, rock type, geomorphic type | $p < 0.01$, $p < 0.05$ |
| [31] | May 2018 | Northwest China | agricultural soils | 62 | Geo-detector | Cd, Cr, Cu, Ni, Pb, Ti, Zn | distance from industrial enterprises, altitude, soil pH, distance from major roads | $p < 0.001$, $p < 0.01$, $p < 0.05$ |
| [32] | 2015 | Zhejiang, China | agricultural soils | 1928 | GeogDetector | As, Cd, Cr, Pb, Hg | soil parent materials, farmland type, industrial production, number of cars (1/1000), fertilizer use, pesticide use | $p < 0.01$ |
| [34] | First half of 2013 | Hechi, Guangxi | cropland, forestland, grassland, construction land | 513 | SOM clustering, Geographical Detector | As, Cd, Cr, Hg, Pb | rivers, factories, ore zones | $p < 0.05$ |
| [35] | November 2016 | Shenzhen, Guangdong | urban soils | 221 | Geographical Detector | As, Pb | original bedrock, subsequent pedogenesis, industrial wastes, vehicle emissions, household garbage | $p < 0.05$ |

Based on the same location, the examined studies included two aspects, named related study I [17] and related study II [43], respectively. The results of the comparative analysis

are shown in Table 7. Table 7 shows that the multi-object identification results between heavy metals and influencing factors included that of single-object identification in this study. Compared with related studies, the differences in identification results might be caused by different sampling times, locations, and identification methods for this study and related studies, as depicted in Table 7. Furthermore, we compared our study with the related study I over the same study area and heavy metals, as shown in Figure 8. This figure shows that the results derived at the single-object level using the geographical detector are significantly different from that of the existing method using PCA. In general, the proposed method at the multi-object level identified more influencing factors than the method based on the single-object level and the PCA method. Another interesting finding is that the derived results at the multi-object level refer to the combination of the results from the single-object method and the PCA, except for the variable of Pb. More specifically, (1) for variables of As and Zn, more influencing factors identified by the multi-object method than either the single-object identification or the PCA method; (2) for variables of Cd, Hg, and Cu, the influencing factors identified by the multi-object method were equal to the sum of results of the single object method and the PCA method. According to PCA results for six heavy metals in Table 3 and identification results in Table 4, the potential reason was that Pb was secondarily represented in the second component, and the second component was associated with soil properties, due to selecting only the maximum principal component for each heavy metal in this study. Therefore, the identification method developed at different levels in this study can identify much more influencing factors of heavy metals than frequently used methods, particularly for the multi-object identification method.

**Table 7.** The comparative results between this study and related studies.

| Studies | Time | Location | Samples | Methods | Metals | Influencing Factors |
|---|---|---|---|---|---|---|
| Our study (single object) | Autumn 2007 | Shunyi District | 329 | Geographical detector | As, Cd, Cu Cd, Hg, Pb | DEM Annual deposition flux |
| Our study (multi-object) | Autumn 2007 | Shunyi District | 329 | PCA; Geographical detector | Cd, Cu, Zn<br><br>As<br><br>Hg, Pb | DEM, land use, annual deposition flux<br>Soil type and soil texture, DEM<br>Annual deposition flux |
| Related study I [17] | August to November 2009 | Shunyi District | 412 | PCA | Cd, Cu, Zn As, Pb Hg | Agricultural practices Soil parent materials Atmospheric deposition |
| Related study II [43] | - | Beijing | 773 | PCA; CA | As, Cr, Ni<br><br>Cd, Cu, Pb, Zn<br><br>Pb, Zn, Cu | Pedogenic factors Anthropogenic and soil parent factors Traffic and smelting |

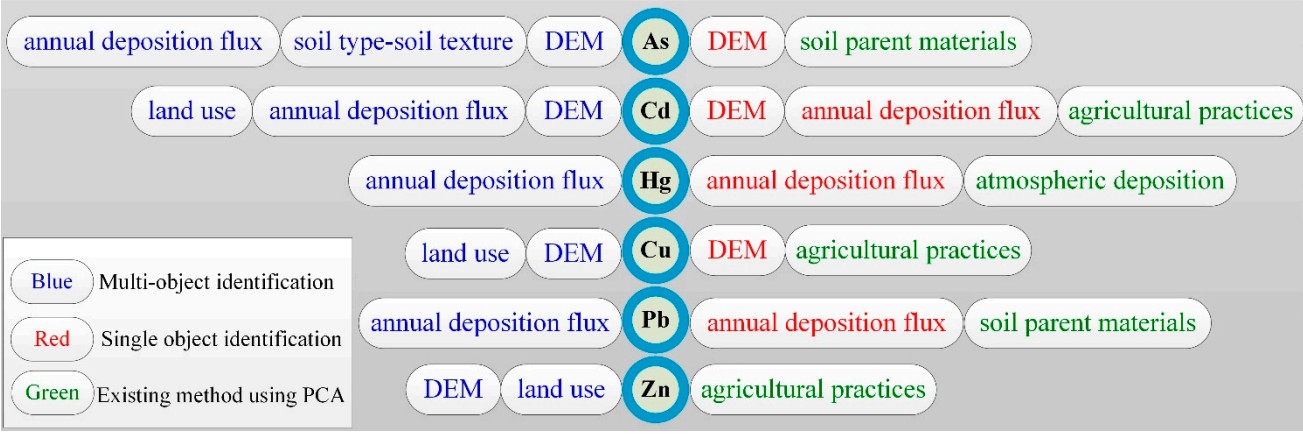

**Figure 8.** The identification results for this study and the related study I [17].

*4.2. Uncertainty and Proper Use of Identification Method Developed in This Study*

Although the proposed method achieved satisfactory results, further improvement can be conducted by interpolating the results of the atmospheric deposition samples and the stratification number of influencing factors, particularly for numerical variables. More specifically, the IDW interpolation results of dry and wet atmospheric deposition may lead to additional uncertainty to some extent in this study. An ideal IDW interpolation result is to obtain each point's actual value, making it challenging to describe high concentration when overall concentration is low [7]. Regarding influencing factors of heavy metals, the number of stratifications was determined according to prior knowledge in this study. By doing so, the number of stratifications should be determined in detail to express the spatial variability of heavy metals when using a geographical detector [42]. Furthermore, to select a geographical detector for identifying influencing factors, the significance level of *q* values needs to be specified, e.g., the significance at $p < 0.05$ level was used in this study.

We also provided suggestions for the proper use of the identification method developed in this study. Variation in data sources, study areas, and research targets should be considered for future implementation when applying the proposed method at different levels. For the cases with fewer variables, we recommend using the single-object method for identifying influencing factors of heavy metals; otherwise, we suggest using the proposed multi-object identification method.

## 5. Conclusions

This paper presented an identification method based on a geographical detector to identify natural and anthropogenic factors of six heavy metals (i.e., As, Cd, Hg, Cu, Pb, Zn) at single-object and multi-object levels in Shunyi District, China. The results suggested that Cd, Cu, and Zn concentrations were mainly influenced by DEM and land use factors, while annual deposition fluxes were the main factors of Hg, Cd, and Pb concentrations. Moreover, the concentration of As was primarily influenced by soil properties, DEM, and annual deposition flux. The multi-object identification results between heavy metals and influencing factors included that of single-object identification in this study. Compared with the frequently used PCA and CA methods, the identification method developed at different levels can identify much more influencing factors of heavy metals, particularly using the multi-object identification method. Due to its promising performance, the method developed at different levels can be widely employed for soil protection and pollution restoration.

Nevertheless, we expect that future work can be improved from the following two aspects: (1) to analyze whether the influencing factors of agricultural soil heavy metals operate independently or interconnect, and to quantify the interaction effect; (2) to identify and monitor high pollution risk areas in combination with the background values of heavy metals in agricultural soils.

**Author Contributions:** Conceptualization, S.D.; methodology, S.D. and B.G.; software, S.D. and H.G.; validation, S.D., H.G., and M.L.; writing—original draft preparation, S.D.; writing—review and editing, B.G., Y.P., and M.L.; funding acquisition, S.D. and Y.P. All authors have read and agreed to the published version of the manuscript.

**Funding:** This research was funded by the Beijing Natural Science Foundation (Grant Number 8192015) and the National Natural Science Foundation of China (Grant Number 41801276).

**Acknowledgments:** We thank Yanan Wu and Dongyue Zhang from National Engineering Research Center for Information Technology in Agriculture for data processing.

**Conflicts of Interest:** The authors declare no conflict of interest.

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
