# Peer review of "Identifying Influencing Factors of Agricultural Soil Heavy Metals Using a Geographical Detector: A Case Study in Shunyi District, China"

_land, doi:10.3390/land10101010_

Round 1

Reviewer 1 Report

The authors have presented an interesting manuscript that is related to the journal topic and related to the current problematics of soil pollution and agriculture role. There are some aspects that must be solved before accepting the paper. Following, I mentioned the most relevant aspect to be improved:

Major issues:

In order to evaluate the novelty of this study properly, a related work section (or a subsection of background in the introduction) must be included. In this section (or subsection), the authors have to analyze in-depth the existing contributions to the topic of “Soil Heavy Metals” and “Geographical Detector”. After a fast check-in scholar google, a lot of papers are found on this topic. The authors must clarify their contributions and the main differences between their work and the rest of the published papers. The authors must cite the relevant contributions in this section, explaining their studied area, the statistical method, etc. After describing the related work, the authors have to identify the gap in the existing solutions they will cover with their paper.

In the discussion, before starting the comparison of their results with other published papers, authors should discuss their findings in terms of the influence of parameters on heavy metals. The authors must discuss why different elements are described by the different factors and the most probable explanation for this phenomenon. It will be useful if authors cite other papers that support the explanations. In this point, the mention of agriculture and its effects is essential to justify the title of the paper. Authors must explain if the agricultural practices are the origin of the pollution and the effect of the pollutants on the future development of agriculture.

In Table 5, more contributions have to been added in order to provide a more comprehensive discussion. If authors want to focus one table on the contributions of the same location, another table can be added. In this table authors must include the contributions cited in the related work (as much as possible), including all the parameters of Table 5, and if possible, include the minimum and maximum concentration of each component, the p-value of encountered correlations, etc.

Minor issues:

- I suggest avoiding using in the keywords the words used in the title. Therefore, I propose the authors change the words “heavy metals”, “influencing factor”, and “geographical detector” by synonyms or other keywords.

- In the abstract, the authors should include the relevant p-values for their affirmations.

-  In Table 5, the references of Related Study (i) and (ii) should be included.

- At the end of the conclusions, the authors have to include a new paragraph detailing the future work related to their findings.

Reviewer 2 Report

The manuscript (MS) “ Identifying Influencing Factors of Agricultural Soil Heavy Metals Using Geographical Detector: A Case Study in Shunyi District, China ”, contributes to  geographical detector to identify influencing factors of agricultural soil heavy metals from natural and anthropogenic aspects. The topic fits the aims and scope of the Land.

This is a useful study. A lot of work has been done, and an impressive number of soil samples have been collected and analyzed.In my point of view, the manuscript needs in some transformation.

  1. For clarity, I recommend building maps of the distribution of each metal. It would be useful for visualization the data.
  2. Usually, the PCA is used in combination with other methods. I recommend to use the correlation method as well as the cluster analysis method.
  3. In the discussion of the results, it is necessary to pay more attention to explaining the possible reasons for the influence of each factor on each metal, to provide links to other studies.

Round 2

Reviewer 1 Report

The authors have correctly addressed all the comments, and the paper is now ready to be published.

This manuscript is a resubmission of an earlier submission. The following is a list of the peer review reports and author responses from that submission.